# Characterization of Pectin Oligosaccharides Obtained from Citrus Peel Pectin

Diana Pasarin [1], Andra-Ionela Ghizdareanu [1,2,*], Florina Teodorescu [3,*], Camelia Rovinaru [1] and Alexandra Banu [2]

[1] National Research and Development Institute for Chemistry and Petrochemistry-ICECHIM, 202 Splaiul Independentei, 060021 Bucharest, Romania
[2] Faculty of Material Science and Engineering, University Politehnica of Bucharest, 313 Splaiul Independentei, 060042 Bucharest, Romania
[3] "C.D. Nenitzescu" Institute of Organic and Supramolecular Chemistry of the Romanian Academy, 202B Splaiul Independentei, 060021 Bucharest, Romania
* Correspondence: ghizdareanuandra@gmail.com (A.-I.G.); florina.teodorescu@gmail.com (F.T.)

**Abstract:** This study aims to characterize the pectic oligosaccharides (POSs) generated from enzymatically hydrolyzed citrus peel pectin using a selected enzyme. Pectinex Ultra AFP was used to depolymerize citrus peel pectin into POSs. The POSs were analyzed using high-performance liquid chromatography (HPLC) and liquid chromatography coupled with a mass spectrometer (LC/MS) methodology to determine the composition of monosaccharides and the average molar mass distribution based on the retention time. The identified fractions were predominantly neutral sugars (rhamnose, glucose, and galactose) and acidic sugars (galacturonic acid), with corresponding mole percentages of 8.67%, 10.28%, 74.33%, and 6.72%, respectively. The degree of polymerization (DP) was in the range of DP3–DP8, containing three (trimers) to eight (octamers) monomeric units. The low DP indicates an advanced degree of enzymatic hydrolysis of pectin up to the level of pectic POSs.

**Keywords:** citrus peel wastes; enzymes; monosaccharides; pectin; pectic oligosaccharidesr





## 1. Introduction

Pectin is a complex polysaccharide that is widely distributed in the cell walls and middle lamina of higher plants. It is composed of galacturonic acid residues, and its structural complexity varies depending on the plant source and extraction method used. The unique structural properties of pectin include its high degree of branching, the heterogeneous distribution of galacturonic acid residues, and the presence of neutral sugars and acetyl groups in its side chains. The biochemical properties of pectin are influenced by its structure and include: its ability to form gels in the presence of sugar and acid, its function as a stabilizer in emulsions, and its role as a dietary fiber in the human diet. The diverse applications of pectin are attributed to its unique structural and biochemical properties [1,2]. Pectin has various industrial applications, such as in the food and pharmaceutical industries, as well as in the development of edible films, plasticizers, paper substitutes, and foams. One of the most promising applications of pectin is in the production of POSs, which are short chains of monosaccharides. POSs derived from pectin, known as pectic POSs, have been shown to have various bioactive properties, including antioxidant, prebiotic, and immunomodulatory properties. POSs can be produced from pectin by enzymatic hydrolysis, which selectively cleaves the glycosidic bonds of pectin into shorter-chain POSs [3]. Citrus peels are a rich source of pectin and have the potential to be a sustainable and cost-effective feedstock for pectin production. However, the structural complexity of citrus pectin can make its extraction and processing challenging, and there is a need for efficient and cost-effective methods to produce high-quality pectin products [4]. When used in the food sector, pectin films offer great hardness and good adhesive properties,

acting as a natural barrier for the exchange of moisture, gases, lipids, and volatiles between the external and internal environment. This contributes to the prevention of food microbial contamination [5]. However, pectin films also have some disadvantages, such as stiffness, brittleness, and susceptibility to water [6].

Pectin has been shown to possess a variety of health functions that are the subject of extensive research. These functions include bio-sorption, immunomodulation, anti-proliferation activity, antioxidant, antimicrobial, hypocholesterolemic, and hypoglycaemic properties [7–9]. These properties make pectin a valuable resource for various applications, including the food, pharmaceutical, and biomedical industries. For example, pectin has been investigated for its potential use as a dietary fiber additive and as a natural alternative to synthetic food additives and preservatives [10]. In addition, pectin-based materials have been developed for drug delivery, tissue engineering, and wound healing due to their biocompatibility, biodegradability, and functional properties [11,12]. Pectin is a unique polysaccharide that exhibits remarkable bioactivity and biocompatibility, making it an attractive material for various food, pharmaceutical, and biomedical applications. The bioactivity of pectin refers to its ability to interact with biological systems and modulate their functions, such as inhibiting enzyme activities, binding to toxins, and promoting cell growth and differentiation [8,9]. On the other hand, biocompatibility refers to the ability of a material to interact with living tissue without causing an undesirable reaction, such as inflammation or toxicity. Pectin has been shown to have excellent biocompatibility because it is a natural, nontoxic, and biodegradable polysaccharide that can be metabolized by the gut microbiota [7,12]. In addition, pectin is renewable, inexpensive, and easy to modify, which enables the development of novel materials with tailored properties and functions [10,11].

Citrus fruits are a group of fruit-bearing trees and shrubs that belong to the Rutaceae family. They are widely cultivated in tropical and subtropical regions and are consumed worldwide. Citrus peels, which are a byproduct of the citrus industry, contain high levels of pectin, a complex carbohydrate that is commonly used as a gelling agent in the food industry. Citrus peels and apple pulp are the major commercial sources of pectin, but it can also be obtained from other alternative sources such as sugar beet pulp, rapeseed cake, mango or peach peels, pumpkin pulp, olive wastes, and onion skins [13]. According to a recent report by the United States Department of Agriculture, global production of oranges was expected to rise in 2021 by 7% to 49.4 million metric tons. Orange production in the EU has also increased [14].

A significant portion of this production is used for juice extraction, resulting in a considerable amount of waste that has a polluting effect. However, this waste can be exploited as a source of value-added products. Carbohydrates are the most abundant compounds in the pericarp of citrus fruits [15] and are represented by cellulose, hemicellulose, and pectin [16]. Pectin accounts for about 20–30% of the dry weight [17].

The chemical composition and structure of pectin have been extensively studied over the years, and while there is a general consensus on the main characteristics of pectin, there are still some aspects that are the subject of ongoing discussion and research. One of the main points of debate regarding pectin is its chemical composition. Pectin is composed of a number of different sugar molecules, including galacturonic acid, rhamnose, arabinose, and galactose. The exact arrangement of these sugars in the pectin molecule can vary, which can affect its properties and functionality. Another aspect of pectin that is still studied is its structure. Pectin molecules can be highly branched, and the degree of branching can vary depending on the source of the pectin. In addition, pectin molecules can form cross-links with other polysaccharides and proteins in the cell wall, further complicating its structure. The composition of pectin varies depending on the plant source, the degree of maturity of the raw material, the methods and conditions of extraction, and various environmental factors, which makes the characterization of the pectin structure difficult [18]. Currently, the consensus is that pectin consists of three characteristic structural domains (types of polysaccharides). The first characteristic structural domain, homogalacturonan (HG), or

the smooth region, is the most abundant component, accounting for at least 65% of all pectins in cell walls. HG is a linear polymer formed from 1,4-linked D-galacturonic acid (GalA), which is partially methyl-esterified at the C-6 carboxyl and may be O-acetylated at the O-2 or O-3 positions, depending on the source. This type of esterification is crucial for the biochemical, physical, and functional properties of pectin [19,20]. Cho et al. [21] have demonstrated two methods for converting high methoxyl pectin to low methoxyl pectin: chemical de-esterification by alkali and enzymatic treatment by pectin methyl esterase. Chemical de-esterification of high methoxyl pectin usually uses acids, alkalis, and alcoholic/aqueous solutions to produce low methoxyl pectin. Pectins from apples, citrus fruits, and sugar beets have the backbone of HG, which comprises approximately 72–100 D-GalA residues [22]. The second characteristic structural domain, rhamnogalacturonan I (RG -I) or the hairy region, is a backbone of repeating and alternating sequences of [(→2)-$\alpha$-L-Rhap-(1→4)-$\alpha$-D-GalAp-(1→)] disaccharide residues that constitute up to 7–14% of pectin. Some of the $\alpha$-L-rhamnose residues (Rha) are replaced at O-4 and rarely at O-3 by neutral side chains containing galactose (Gal) and/or arabinose (Ara) residues, such as $\beta$-D-galactans, (1→5)-$\alpha$-L-arabinans, and arabinogalactans with over 15 different linkages [23,24]. The third characteristic structural domain, the rhamnogalacturonan II (RG-II), consists of a homogalacturonan backbone with 7 or 10 $\alpha$-(1→4)-D-GalpA units and four different structural side chains, attached to the C-2 or C-3 positions of GalA residues, including rare monomers, such as apiose, 2-O-methyl fucose, 3-carboxy-5-deoxy-L-xylose (aceric acid), 2-keto-3-deoxy-D-manno-octulosonic acid, and 3-deoxy-D-lyxo-heptulosaric acid [25–28].

Depending on the raw material, the components of pectin can vary from a standpoint. The abundance of each domain, as well as the molecular mass (MS) of pectin, degrees of acetylation and methylation, contents of acidic (GalA), and neutral sugar residues (Ara, Rha, Gal) show great natural variability between different pectin sources. Pectin from sugar beet pulp has shorter HG chains and is more abundant in RG-I than pectin in citrus peel or apple pulp. Additionally, pectin from sugar beet and spinach contains significant amounts of esterified ferulic acid, which influences antioxidant activity [29]. These differences influence the properties of pectin in different raw materials, as well as their applications. Pectin demonstrates the capability to form complexes with other natural compounds, which makes it useful for designing food products [30].

The proportions and organization of the structural elements in different pectin samples are assumed to have a notable impact on their water-solubility, arrangement, and functional characteristics, including their potential to thicken, gel, form layers, emulsify, generate chains interlinks, and collaborate with other biomolecules as film forming products [12].

Partial enzymatic hydrolysis or other processes [21] can lead to the depolymerization of purified pectin or pectin present in suitable raw materials to obtain pectic POSs, which can contain a variety of diverse structural elements present in pectin, including: oligogalacturonides (OGalA), alpha-galactooligosaccharides (GalOS), arabinooligosaccharides (AraOS), rhamnogalacturonoligosaccharides (RhaGalAOS), xylooligogalacturonides (XylOGalA), and arabinogalactooligosaccharides (AraGalOS) [31]. POSs consist of 3–10 or up to 20 monosaccharide residues or even disaccharides, with low molecular weight. The POS is formed by the joining of monosaccharide units via glycosidic bonds between the anomeric carbon of one sugar and any one of the unmodified hydroxyl groups in another monosaccharide [32]. POSs are also commonly bound to lipids and amino acids by way of *O*-glycosidic and *N*-glycosidic bonds to produce glycolipids and glycoproteins according to some authors [33]. Various analytical techniques have been used for determining the structure and chemical composition of pectin POSs originating from pectin from different sources. The predominant monomeric units are D-GalpA, L-Ara, and D-Gal, while L-Rha and D-Xyl are present in small quantities [19,34–36]. POSs play an important role in cellular communication, inhibition of pathogen adhesion in the gastrointestinal tract, and stimulation of apoptosis of human colonic adenocarcinoma cells [20].

POSs are able to reach the distal part of the colon where they can suppress the MAPK signaling pathway, promoting the apoptosis of colon cancer cells. POSs with a certain length of the chain produced from the homogalacturonan and rhamnogalacturonan domains [37] have been regarded as emerging prebiotics with superior ability to increase the probiotic flora in the gastrointestinal tract, particularly species such as *Faecalibacterium prausnitzii*, *Roseburia intestinalis*, *Lactobacillus*, *Eubacterium*, and *Bifidobacteria* [23]. Additionally, other uses for pectin, such as pectin films, can be produced using various methods, such as casting, extrusion, injection, and coating. Pectin films from apple, carrot, or hibiscus, containing carvacrol or cinnamaldehyde, were tested for their antibacterial activity against *Listeria monocytogenes* on contaminated ham and sausage. The results showed that the pectin film with carvacrol had higher antimicrobial activity than the pectin film with cinnamaldehyde and that the membrane developed was more effective on ham than on sausages. Furthermore, the use of edible apple pectin films resulted in greater inactivation of this pathogenic microorganism than carrot or hibiscus pectin films [38].

Starting from citrus peel pectin, mixtures of POSs containing various types of oligomers are obtained. Characterizing these oligomers is the main challenge, so various analytical techniques have been used such as high-performance size exclusion chromatography (HPSEC), [39] high-performance anion exchange chromatography (HPAEC-PAD) [23,29] mass spectrometry [40], hydrophilic interaction liquid chromatography-electrospray ionization mass spectrometry (HILIC-ESI/MS) [27], HILIC coupled to traveling-wave ion mobility mass spectrometry (TWIMMS) [26], gel permeation chromatography combined with multi-angle light scattering (GPC-RI-MALS) [28], Fourier transform infrared spectroscopy (FTIR), and mass spectroscopy [21]. To the best of our knowledge, there is no information available regarding the characterization of citrus peel POSs using liquid chromatography coupled with a MS analyzer and time-of-flight detector, model 6224 TOF LC/2MS (Agilent Technologies, Agilent Technologies, Inc., Santa Clara, CA, USA).

Considering the above, the aim of this work is to produce a concentrate rich in POSs from citrus peel pectin through enzymatic treatment with the commercial enzyme Pectinex Ultra AFP. The resulting mixture of POSs will then be analyzed to determine the hydrolysis pattern of citrus peel pectin under the action of this enzyme.

## 2. Materials and Methods

### 2.1. Chemicals

Citrus peel pectin with a high degree of methoxylation (>70.0%) was kindly provided by Döhler, Germany. Standard monosaccharides glucose, mannose, rhamnose, arabinose, D-galactose, galacturonic acid, and xylose were procured from Sigma Aldrich and used as received, except for D-galactose and galacturonic acid, which were recrystallized to purify them. D-galactose was dissolved in hot water, cooled to 0 °C and poured into absolute ethanol. Crystallization occurred within minutes after vigorous stirring. Galacturonic acid was recrystallized from 95% ethanol and dried in a vacuum desiccator over $P_2O_5$. Trifluoroacetic acid (TFA) was purchased from Alfa Aesar, and 3-Methyl-1-phenyl-2-pyrazoline-5-one (PMP) was supplied by Acros Organics.

The commercial pectinolytic enzyme Pectinex Ultra AFP was kindly supplied by Novozymes (Bagsvaerd, Denmark). The liquid enzyme Pectinex®Ultra AFP, produced from *Aspergillus aculeatus* and *A. niger*, is a blend of pectin lyase and polygalacturonase with activities of 10,000 Units/mL. Pectin lyase (E.C. 4.2.2.10) catalyzes α-(1,4) linkages by trans-elimination, resulting in GalA units with an unsaturated bond between C4 and C5 at the non-reducing end of the GalA formed [41]. Polygalacturonase (E.C. 3.2.1.5) hydrolyses α-(1–4)-glycosidic linkages between D-GalA from homogalacturonan. Hydrochloric acid was provided by Scharlau (Scharlab SL, Scharlau Leathergoods, Barcelona, Spain), and sodium hydroxide by SC Chimreactiv SRL (Neamt, Romania). All other analytical chemicals were purchased from Merck, Germany.

### 2.2. Enzymatic Hydrolysis of Citrus Peel Pectin

In the current study, citrus peel POSs were obtained by enzymatic hydrolysis using Pectinex Ultra AFP, following the method described by Sabater C. [42], with modifications. Briefly, 2 g of citrus peel pectin samples were dissolved in water at a solid-to-solvent ratio of 1:25 (*w/v*), in 250 mL working volume Erlenmeyer flasks. The pH of the mixture was adjusted to 4.5 using NaOH 1 N. The mixture was gently stirred on an orbital shaker for 30 min to allow for equilibration. Enzyme volumes of 5 μL (C1), 12.5 μL (C2), and 100 μL/g of pectin (C3) were added to the mixture. The enzymatic preparations had no measurable activity other than their main activity. The mixtures were incubated for 1.5 h and 2 h at 45 °C and 53 °C and homogenized in an orbital shaker at 170 rpm to increase the surface area for an efficient enzyme treatment. The hydrolysis parameters (temperature, time, and enzyme concentration) were optimized by The Orthogonal Array Testing Strategy (OATS). At the end of the reaction period, the enzyme was inactivated by thermal treatment at 90 °C for 10 min. After centrifugation at $5433 \times g$ for 15 min, 10 mL aliquots were taken from the resulting supernatants containing the pectic POSs mixture, extracted and filtered through a 0.45 μm membrane filter, and chemically analyzed. The remaining supernatant was concentrated using a rotavapor and lyophilized. The experiment was performed in duplicate.

The yield of POSs was calculated according to Equation (1):

$$\text{POSs extraction yield } = \frac{\text{quantity of dried OP}}{\text{quantity of pectin}} \times 100, \tag{1}$$

### 2.3. LC/MS Analysis of Hydrolyzed Citrus Peel Pectin

The enzymatic hydrolysate of citrus peel pectin was characterized for molecular mass using HPLC coupled with a MS and a time-of-flight detector, (Agilent Technologies, model 6224 TOF LC/MS). The system was calibrated using a mass reference solution (ESI-L Low concentration tunning mix, code G1969-85000, Agilent Technologies). MS detection was performed using an orthogonal TOF-MS coupled to an ESI source, with double spray needles for continuous infusion of the reference mass solution. The drying gas, heated to 350 °C, with a flow rate of 9.0 L/min nitrogen at a pressure of 40 psig (gauge pressure), was used to dissolve the solution drops. The spray was induced with a capillary voltage of 3500 V, and the fragmentation voltage was 100 V.

The analyte solution (5 μL), filtered through 0.22 μm PTFE syringe filters, was injected directly into the MS without chromatographic separation. The mobile phase used for taking and transferring the sample was 80% acetonitrile with 0.01% trifluoroacetic acid, and a quaternary HPLC pump with a flow rate of 0.3 mL/min was employed. Data acquisition and qualitative processing were performed using Mass Hunter software, version B.04.00. The data acquisition range was set to 600–2000 *m/z*, with 9894 scans and a scan rate of 1 scan/s.

The length of the pectin polymer chain is quantified [43] by the DP, which was determined using Equation (2):

$$\text{DP} = \frac{\text{Mn}}{194.14 - 18.02}, \tag{2}$$

where Mn represents the number-average molecular mass; 194.14 is the molecular weight of one GalA unit; 18.02 is the molecular mass of water.

### 2.4. HPLC Analysis of Hydrolyzed Citrus Peel Pectin

The identification of citrus peel pectic POSs was carried out by the aforementioned HPLC chromatographic method. The POSs solution (citrus peel pectic POSs) was concentrated under reduced pressure using a Heidolph Hei-VAP Core HL G3 rotary evaporator at 50 °C, 1 bar, 130 rpm. A sample of 17 mg of concentrated POSs was added to a round-bottom flask, followed by the addition of 2 mL of 3 M TFA solution. The flask was sealed with argon and placed in an oil bath heated at 110 °C for 4 h. The reaction mixture was then

neutralized using 1 M NaOH solution and used for further derivatization. The monosaccharides derivatization was carried out according to method described by Honda et al. [44] method. Briefly, 0.5 mol of monosaccharides was dissolved in 5 mL of 0.3 M NaOH solution, followed by the addition of 10 mL of 0.25 M PMP solution. The mixture was allowed to react for 2 h at 70 °C, then cooled in an ice bath and neutralized using 0.3 M HCl solution. The solution was extracted with $CHCl_3$ three times, and the aqueous phase was used for HPLC analysis. The hydrolysate samples were centrifuged, filtered through 0.22 μm membranes, and then analyzed by HPLC. The analysis was conducted using an Agilent 1200 Chromatograph equipped with an Agilent Zorbax SB-C18 (250 × 4.6) column. The mobile phase A was 0.1 M phosphate buffer with a pH of 6.7, and the mobile phase B was acetonitrile, with a 83:17 volume ratio between the two. The elution was performed at a flow rate of 1.0 mL/min at 35 °C, using a 10 μL injection volume and a detection wavelength of 254 nm.

### 2.5. Statistical Analysis

Statistical analysis of data was performed with Minitab Version 18. All results were expressed as mean values ± standard deviation based on at least two measurements ($n = 2$), and a one-way analysis of variance (ANOVA) was conducted using a general linear model procedure. Comparison procedures assuming equal variances were carried out by Tukey's test for a statistically significant level of $p < 0.05$.

## 3. Results and Discussion

The citrus peel pectin was enzymatically hydrolyzed and assayed to determine the composition of monosaccharides and analyze their molecular mass. After enzymatic hydrolysis of 2 g of citrus peel pectin, 1.22 g of pectic POSs were obtained. The extraction yield of the POSs was 61%.

### 3.1. HPLC Analysis of Hydrolyzed Citrus Peel Pectin

Chromatographic analysis of hydrolyzed citrus peel pectin with different enzyme concentrations showed that the highest degree of pectin fragmentation was obtained at a concentration of 12.5 μL/g of pectin (C2). When comparing the HPLC chromatograms of citrus peel pectin and hydrolyzed citrus peel pectin, at the same concentration of the analyzed samples (Figure 1), it can be observed that the pectin peak area decreases significantly, indicating that pectin fragmentation was nearly complete.

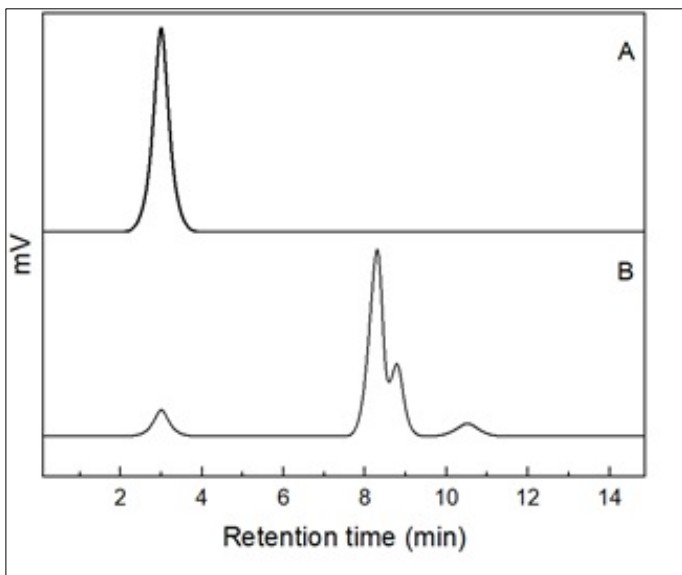

**Figure 1.** Chromatogram of HPLC analysis of citrus peel pectin (**A**) and hydrolyzed citrus peel pectin (**B**) with Pectinex Ultra AFP in a concentration of 1.25 μL/g.

Three independent variables were used to optimize the enzymatic hydrolysis of citrus peel pectin: enzyme dose, temperature, and extraction time. The results obtained regarding the degree of fragmentation (dependent variable) of the tested samples were statistically analyzed.

Figure 2 shows an interval plot of enzyme concentration comparing confidence intervals. It can be observed that the degree of fragmentation varies for each enzyme concentration when the other parameters (temperature and time) change.

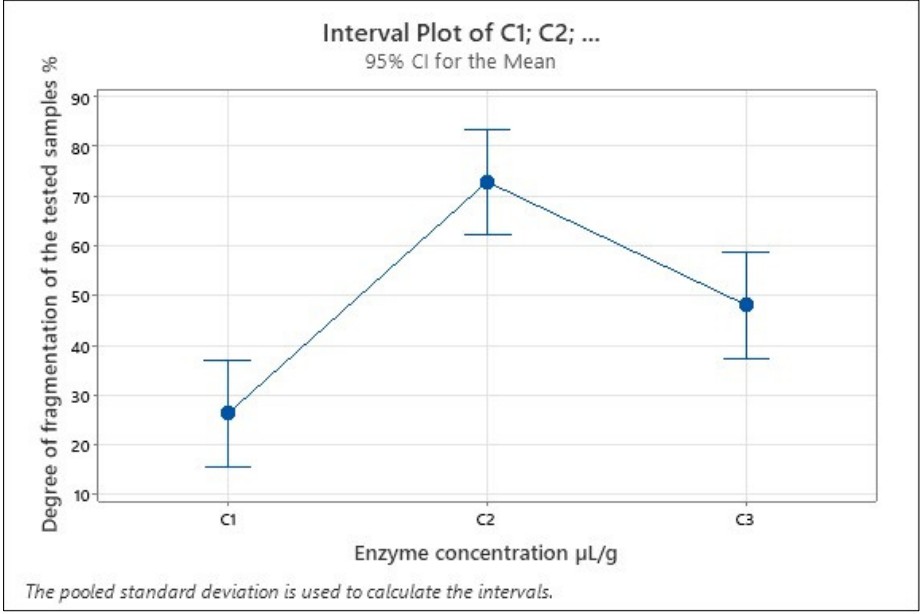

**Figure 2.** Degree of fragmentation for each concentration when the other parameters change (temperature and time).

The combined standard deviation among the concentrations was used to calculate the intervals. To determine which means differ from the rest in a set of means, Tukey's multiple comparison tests were applied (Table 1).

**Table 1.** Comparison procedures assuming equal variances by Tukey.

| | | Degree of Fragmentation of the Tested Samples | | |
|---|---|---|---|---|
| **Temperature (°C)** | **Time (h)** | **Enzyme Concentration, µL/g** | | |
| | | **0.5** | **12.5** | **100** |
| 45 | 1 | 24.74 ± 0.60 [a] | 81.72 ± 0.15 [b] | 40.00 ± 0.45 [c] |
| 45 | 1.5 | 28.00 ± 0.21 [a] | 82.69 ± 0.41 [b] | 45.00 ± 0.17 [c] |
| 53 | 1 | 16.86 ± 0.24 [a] | 61.48 ± 0.50 [b] | 45.00 ± 0.14 [c] |
| 53 | 1.5 | 35.00 ± 0.14 [a] | 65.00 ± 0.17 [b] | 61.48 ± 0.22 [c] |

Results are expressed as the mean ± standard deviation ($n = 2$). The values followed by different letters indicate significant differences ($p < 0.05$), the $p$-value is 0.024.

According to the results obtained, the highest degree of pectin fragmentation was obtained at the enzyme concentration of 12.5 µL/g, a temperature of 45 °C, and an extraction time of 1.5 h.

This hydrolyzed sample was further analyzed to determine the composition of monosaccharides. To evaluate the qualitative and quantitative monosaccharide composition of citrus peel pectin POS, each monosaccharide standard was analyzed, and calibration curves were generated by analyzing five points in the range 0.1–1.8 mM. The HPLC analysis of the hydrolyzed sample revealed the presence of four monosaccharides, as shown in Figure 3.

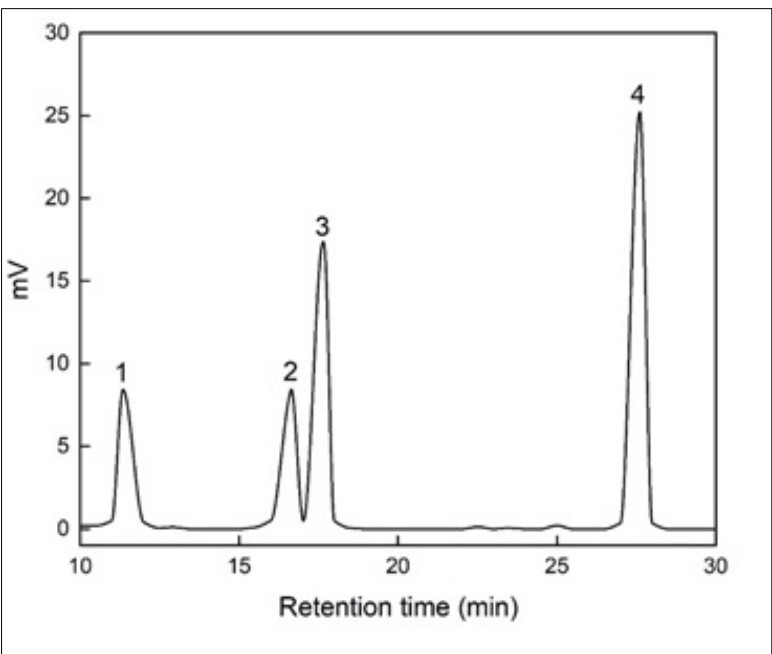

**Figure 3.** The monosaccharides composition of the analyzed sample: 1. rhamnose; 2. galacturonic acid; 3. glucose; 4. galactose.

The monosaccharide components were determined by comparing the retention time of standard monosaccharides with the chromatogram of the mixture under the same conditions. The peaks were identified, and the predominant composition of monosaccharides in the citrus peel pectin was galacturonic acid (an acidic sugar) and neutral sugars, including rhamnose, galactose, and glucose. Their corresponding mole percentages were calculated based on the chromatographic signals of all identified POS, namely, 6.72%, 8.67%, 10.28%, and 74.33% (mol%), respectively.

**L-rhamnose** is a sugar that requires a longer release time due to its multiple bonds with galacturonic acid [45]. The experimental findings indicated that **galactose** was the major monosaccharide (74.33%, mole/%) obtained after derivatization, suggesting high generation of galacto-oligosaccharides. The amount of galactose in pectin from different sources varied the most compared to other neutral constituents, whose relative weights were constant, which agrees with some reports [46]. It is estimated that pectin contains at least 17 types of monosaccharides in its structure, the majority of which are galacturonic acid, followed by galactose, arabinose, or rhamnose. Galacturonic acid, which is an oxidized form of D-galactose, can be converted to the neutral sugar galactose by a reduction reaction of the carboxyl group. The significant release of galactose during the hydrolysis of citrus peel pectin reflects the importance of RG-I structures, which have a main chain of rhamnose and galacturonic acid with lateral chains of galactose and arabinose, bonded in various ways. As a result, the enzyme Pectinex Ultra AFP has an affinity for the RG-I structural domains.

A significant amount of **glucose** (10.28%), which is not pectic carbohydrate, was found, and it mainly comes from hemicellulose and cellulose polysaccharides, which are extracted during hydrolysis. This result is similar to other studies which have shown that the Pectinex Ultra AFP also has activity on cellulose and hemicellulose [47]. Another explanation could be the presence of xyloglucans [48] or xylogalacturonan [49] associated with pectic material or co-products that are extracted during industrial pectin production.

It is well known that pectin is a polysaccharide rich in **galacturonic acid**, and it would have been expected that it would be the majority compound in pectic sugars. However, a significant small content of galacturonic acid (6.72%) was found in this study. The probable cause could be slower hydrolysis of the more resistant links between galacturonic acids. In

a study presented by Round et al. [50], it was suggested that, during hydrolysis, there is first a breakdown of the RG-I regions, and, later, there is a breakdown of HG regions.

Thibault et al. [13] obtained different rates of sugar residues, including galactose and arabinose, which have more labile bonds, as well as galacturonic acid, which has stronger linkages by acid hydrolysis of pectin isolated from citrus. The solubilization rate of different sugars was in descending order: arabinose > galactose > rhamnose > galacturonic acid. Round et al. [41] study shows how the amounts of monosaccharides found in hydrolyzed (supernatant) and unhydrolyzed (precipitate) pectic material change with hydrolysis time. The method to obtaining the pectic material, whether enzymatic or acid hydrolysis, also influences the sugar composition of the samples. Khodaei et al. [51] reported that the extracted polysaccharides from potato pulp contained a low amount of GalA in the extract (7.9%), which indicates the high rate of HG hydrolysis by β-elimination during the microwave-assisted alkaline extraction.

The monosaccharide composition of the analyzed sample is similar to the results reported by Zhang et al. [28]. They found that the major component of the supernatant was galactose (72.22%), while galacturonic acid was around 11.49%., The values for galactose and galacturonic acid in the precipitate varied between 5.78 to 26.44% and 63.74 to 89.93%, respectively, depending on the hydrolysis workflow. However, the results obtained in this study are not consistent with those reported by Yapo [16]. He found that the GalA content was over 98 mol% of all of the identified and quantified sugars (DP 81-DP117 galacturonic acid residues), and Rha content was less than 0.2 mol%. This indicates that the HG in citrus peel pectin consists almost exclusively of GalA, regardless of its location in the cell wall, in agreement with previous findings [52].

The absence of **arabinose** can be explained by the lack of the active component arabinase in the enzyme Pectinex Ultra AFP, which would normally produce specific hydrolysis of arabinans and release arabinose [42].

The chromatographic analysis confirmed the fragmentation of citrus peel pectin and identified predominantly neutral fractions (rhamnose, glucose, and galactose), as well as acidic sugars (galacturonic acid).

### 3.2. LC/MS Analysis

The molecular mass distribution of the citrus peel pectic POSs was determined by LC/MS. The total ion chromatogram (ICT) and mass spectrum in the selected range were obtained. Signals specific to POSs with molecular mass were extracted from the analyzed complex mixture. The MS spectrum of the samples contained fragments with molecular mass ranging from 628–1738 Da, namely, with low (MS < 700 Da) and medium (700 Da < MS < 3000 Da) molecular mass POSs (as shown in Figure 4).

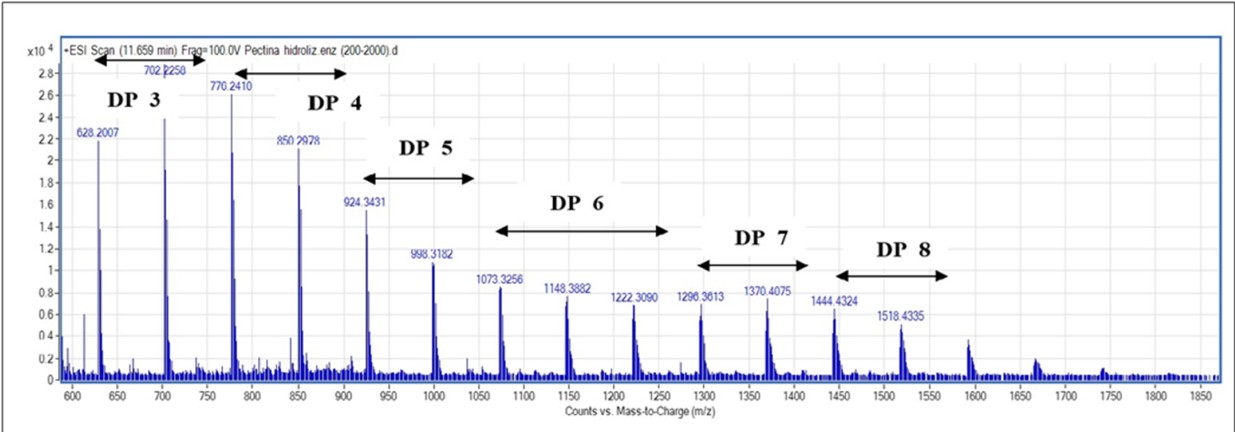

**Figure 4.** MS spectrum of the POS mixture in the range 600–1850 *m/z* (zoom scale) of the enzymatic hydrolysate of citrus peel pectin using Pectinex Ultra AFP.

The average degrees of polymerization at the peaks (DP3-DP8) can be noted.

After two hours of digestion with Pectinex Ultra AFP, more POSs were observed with a regular logarithmic fragmentation of n units of 74 Da, where n max = 15, due to the ionization energy (Figure 1). The symmetry and POS profile are obvious. The presence of relatively small molecular masses (<1.8 kDa) confirms the presence of oligomers, from which the repeated linear detachment of fragments with a molecular mass of 74 Da is observed, possibly from the external ends of the characteristic polygalacturonic chain.

The chromatographic analysis confirmed the presence of different types of oligomers with an average degree of polymerization in the range of DP3-DP8, containing monomeric units from trimers to octamers. The enzymatically hydrolyzed pectic polysaccharides exhibited a specific profile that reflects the complex natural polymer composition with a low degree of polymerization.

Tan et al. [20] reported a normal logarithmic distribution with a 162-Da unit after 72 h of digesting waste citrus pectin using food microorganisms, specifically *Aspergillus niger* 1805. The resulting POSs with DP from 4 to 17 contained glucose monomers, which are considered the main component of fruit peels, including citrus. The chromatographic profiles of hydrolyzed citrus pectin with Viscozyme, as reported by Sabater C. [34], showed an initial molecular mass of pectins of 547 kDa. Chromatographic analysis of enzymatic hydrolysate of apple pectin with Pectinex Ultra AFP, as reported by Wilkowska et al. [53], confirmed the presence of a mixture of POSs with DP in the range of DP1–DP10 after 10 min and DP1-DP7 after 30 min when the pattern of POSs was changed.

Considering the retention times of the commercial standards, the presence of GalA and neutral sugars was assumed, which was further proved by HPLC analysis. MS spectrum of the POS mixture is specific to the complex natural polymer footprint with a low DP.

## 4. Outlook

The findings of this study have practical implications for the food and food packaging industries. The development of POSs from citrus peel pectin offers a sustainable and cost-effective alternative to traditional sources, which could potentially have health benefits and make them attractive for use in foods and food supplements. Future research should focus on developing methods to produce POSs with specific functional properties and conducting clinical trials to confirm their health benefits. In terms of food packaging, the use of pectin-based films offers several advantages, such as biodegradability and low cost. However, the mechanical properties of these films need to be improved to ensure their practical use in the industry.

## 5. Conclusions

Pectin molecules possess enhanced hydrophobic properties due to the presence of methoxyl groups in their structure, making them suitable for a variety of food applications, such as emulsifiers, stabilizers, edible films for food packaging, and bio-composite materials, demonstrating their excellent potential in the food industry. It can be a useful material in many industries due to its bioactivity, biocompatibility, biodegradability, renewability, low cost, and ease of modification.

This study makes an important contribution to the knowledge of citrus peel pectin by demonstrating the efficacy of Pectinex Ultra AFP in the production of POSs from citrus peel pectin, which is a valuable finding, since this specific use of the enzyme has not been previously reported. The POSs were characterized in terms of molecular mass and composition, providing new information to the field. In particular, a mixture of POSs with DP in the DP3-DP8 range was identified, which had not been previously reported. Furthermore, pectin from citrus by-products was found to be high in galactose, indicating the high formation of galacto-oligosaccharides with potential health-promoting properties.

The study highlights the potential of citrus POSs for the development of new and innovative applications in the food and health industries, including their use as a component of food packaging films. However, further research is needed to better understand the

structure–function relationship of pectin POSs and their derivatives to predict their effects and support clinical trials.

**Author Contributions:** Conceptualization, D.P., A.-I.G., F.T. and C.R.; methodology, D.P., A.-I.G., F.T. and C.R.; software, A.-I.G. and F.T.; validation, D.P., A.-I.G., F.T. and C.R.; formal analysis, D.P., A.-I.G., F.T. and C.R.; investigation, D.P., A.-I.G., F.T. and C.R.; resources, D.P., A.-I.G., F.T., C.R. and A.B.; writing—original draft preparation, D.P., A.-I.G., F.T. and C.R.; writing—review and editing, D.P., A.-I.G., F.T. and C.R.; visualization, A.B. All authors have read and agreed to the published version of the manuscript.

**Funding:** This work was supported by a grant of the Romanian Ministry of Research and Innovation, CCCDI-UEFISCDI, project number Cod: PN-III-P3-3.5-EUK-2017-02-0035/contract 129/2019. This work has been funded by the European Social Fund from the Sectoral Operational Programme Human Capital 2014–2020, through the Financial Agreement with the title "Training of PhD students and postdoctoral researchers in order to acquire applied research skills—SMART", Contract no. 13530/16.06.2022—SMIS code: 153734. This research was funded by the Ministry of Research, Innovation and Digitization through Program 1—Development of the national research-development system, Subprogram 1.2-Institutional performance—Projects to finance excellence in RDI, Contract no. 15PFE/2021. This work was carried out through the PN 23.06 Core Program—ChemNewDeal within the National Plan for Research, Development and Innovation 2022–2027, developed with the support of the Ministry of Research, Innovation, and Digitization, project no. PN 23.06.01.01.

**Institutional Review Board Statement:** Not applicable.

**Informed Consent Statement:** Not applicable.

**Data Availability Statement:** No new data were created or analyzed in this study. Data sharing is not applicable to this article.

**Acknowledgments:** The authors thank the Romanian Ministry of Research and Innovation for their support, CCCDI-UEFISCDI, project number Cod: PN-III-P3-3.5-EUK-2017-02-0035/contract 129/2019. The authors also thank European Social Fund from the Sectoral Operational Programme Human Capital 2014–2020 for their funding, through the Financial Agreement with the title "Training of PhD students and postdoctoral researchers in order to acquire applied research skills—SMART", Contract no. 13530/16.06.2022—SMIS code: 153734. The authors thank the funding from the Ministry of Research, Innovation and Digitization through Program 1—Development of the national research-development system, Subprogram 1.2-Institutional Performance—Projects to finance excellence in RDI, Contract no. 15PFE/2021. The authors also want to thank PN 23.06 Core Program—ChemNewDeal within the National Plan for Research, Development and Innovation 2022–2027, developed with the support of the Ministry of Research, Innovation, and Digitization project no. PN 23.06.01.01.

**Conflicts of Interest:** The authors declare that they have no conflict of interest.

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
