# Peer review of "Characterization of Pectin Oligosaccharides Obtained from Citrus Peel Pectin"

_fermentation, doi:10.3390/fermentation9030312_

Round 1
Reviewer 1 Report
1. Line 164-165, “The pH of the mixture was adjusted to 4.5 using NaOH 1 N”, pH was adjust to be acidic pH but why alkaline NaOH solution was used?
2. Add a paragraph practical implications of this study outlining the challenges in the current research, future work, and recommendations, before having short conclusion.
Author Response
Dear reviewer,
We would like to thank the reviewer for the valuable comments/suggestions/recommendations that helped us improve our work.
We inform you that I have thoroughly checked the manuscript and made the necessary modifications based on the recommendations of a native English-speaking colleague.
Please see the attachment.
Best regards,

Reviewer 2 Report
It is a manuscriot that contains little experimental basis, only 3 enzymatic concentrations are tested at 2 temperatures and times. It is also badly written, and the discussion does not reveal any progress in this field. Therefore I do not consider it suitable for publication in this journal.
Introduction. General comment: The introduction is too colloquial and careless, it doesn't seem like a carefully revised work. There are unconnected paragraphs of a line or two, with information that is not clear or explained.
I recommend the authors a thorough review, as an example I highlight some specific comments:
Lines 26-27:The introduction should be a review of the state of the art or background of what is intended to be studied, that is, it is information with references. The first paragraph is too short, it does not contribute much and although it is general information, it is not referenced. I suggest to the authors that they delve into this first sentence.
Line 28: "due to its unique structural and biochemical properties" Which are? Perhaps if the previous paragraph were complete, the information could be linked and would make sense.
Line 38: "which make them indispensable" Again it is indispensable for what or for whom? I understand that there is well known information on pectin, but these paragraphs are overly general and not clear or concise.
Lines 39-40: This phrase is not understood, bioactivity and biocompatibility against or with what?
Line 44: Again a one line paragraph
Line 53: "Pectin chemical composition and structure are still under debate." Maybe I'm not understanding this sentence correctly? I don't think there is any debate about the composition of pectin. In addition, the authors in line 56 say that there is a consensus. Please review the wording of the introduction carefully.
Line 68: "in length [13]. (2) rhamnogalacturonan" Does the sentence start with a (2)?
Line 74 ". (3) rhamnogalacturonan II (RG-II)" I understand that this information is related to the colon used in the previous paragraph. But again punctuation marks are being misused, making the text hard to read.
Lines 127-135: The authors highlight the main bottlenecks in the characterization of pectin, however, according to the last paragraph of the introduction, it is not the object of study and I would have found it very interesting.
Materials and methods
Section 2.2 " The hydrolysis parameters were optimized by The Orthogonal Array Testing Strategy (OATS) (data not shown)" What parameters? Enzyme load, temperature and time? maybe others?
Section 2.4 "The oligosaccharides solution (pectic oligosaccharide) was concentrated under reduced pressure." Please provide more information on the type, conditions or equipment used
Results and discussion: In general, the results are scarce and their discussion does not contribute anything new to the knowledge of citrus pectin. The studied parameters and ranges are not explained and are insufficient. For example, why have these enzymatic loads been decided? Why that jump from 12.5 to 100μL/g? No interesting data is highlighted.
The lack of results could be improved by including the optimization of the hydrolysis, however the authors have decided not to show it.
I suggest authors move the equations to materials and methods
Figure 2 is of very low quality
Author Response

(The authors gave the same response as above.)

Reviewer 3 Report
Despite that in my opinion the paper may be interesting only for the very narrow group of the specialists the manuscript is worth to be published
Author Response

(The authors gave the same response as above.)
